# Document-Level Machine Translation with Large Language Models

**Longyue Wang**[1*]    **Chenyang Lyu**[2*]    **Tianbo Ji**[3*]    **Zhirui Zhang**[1*]
**Dian Yu**[1]    **Shuming Shi**[1]    **Zhaopeng Tu**[1]
[1]Tencent AI Lab    [2]MBZUAI    [3]Dublin City University
{vinnylywang, jackzrzhang, shumingshi, zptu}@tencent.com
chenyang.lyu@mbzuai.ac.ae, tianbo.ji2@mail.dcu.ie

## Abstract

Large language models (LLMs) such as Chat-GPT can produce coherent, cohesive, relevant, and fluent answers for various natural language processing (NLP) tasks. Taking document-level machine translation (MT) as a testbed, this paper provides an in-depth evaluation of LLMs' ability on discourse modeling. The study focuses on three aspects: 1) *Effects of Context-Aware Prompts*, where we investigate the impact of different prompts on document-level translation quality and discourse phenomena; 2) *Comparison of Translation Models*, where we compare the translation performance of Chat-GPT with commercial MT systems and advanced document-level MT methods; 3) *Analysis of Discourse Modelling Abilities*, where we further probe discourse knowledge encoded in LLMs and shed light on impacts of training techniques on discourse modeling. By evaluating on a number of benchmarks, we surprisingly find that LLMs have demonstrated superior performance and show potential to become a new paradigm for document-level translation: 1) leveraging their powerful long-text modeling capabilities, GPT-3.5 and GPT-4 outperform commercial MT systems in terms of human evaluation;[1] 2) GPT-4 demonstrates a stronger ability for probing linguistic knowledge than GPT-3.5. This work highlights the challenges and opportunities of LLMs for MT, which we hope can inspire the future design and evaluation of LLMs.[2]

## 1 Introduction

In the past several years, machine translation (MT) has seen significant advancements with the introduction of pre-trained models such as BERT (Devlin et al., 2018), GPT-2 (Radford et al., 2019), and

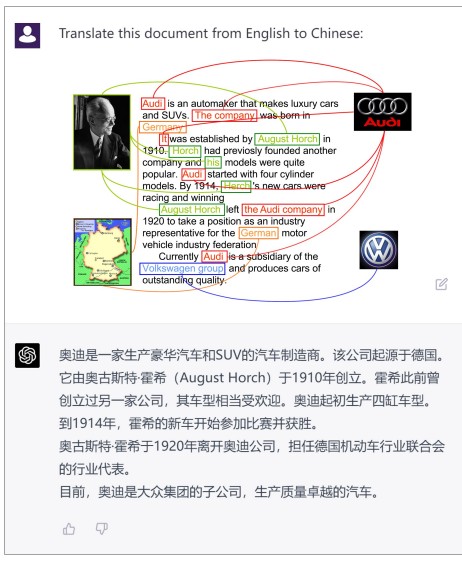

Figure 1: An example of translating a document-level text from English to Chinese using GPT-4 (Date: 2023.03.17). We highlight the discourse phenomena using figures and lines, which are invisible to GPT-4.

T5 (Raffel et al., 2020). These models have demonstrated impressive performance on MT (Zhu et al., 2020; Guo et al., 2020; Xue et al., 2021). However, most of the existing work has focused on sentence-level translation, which can result in translations that lack coherence and context. Recent years have seen a growing interest in *document-level translation*, which is a crucial task that involves translating entire documents (Wang et al., 2017; Bawden et al., 2018; Wang, 2019; Zhang et al., 2022) while modelling specific *discourse phenomena* (Wang et al., 2016; Voita et al., 2018; Wang et al., 2018a,b, 2019; Voita et al., 2019b; Wang et al., 2023b). The most popular large language model (LLM) – Chat-GPT[3] shows the ability of maintaining long-term coherence and consistency in a conversation by conditioning on previous conversational turns. Ad-

---

*Equal contribution.

[1]The protocol employed in this work was approved by the Tencent Institutional Review Board (IRB).

[2]We release our data and annotations at https://github.com/longyuewangdcu/Document-MT-LLM.

[3]https://chat.openai.com. All corresponding results were obtained from GPT-3.5 and GPT-4 in March 2023. The reproducibility is discussed in Section Limitation.

ditionally, the model is trained on a large dialogue dataset, which allows it to learn the patterns and conventions of human communication, further improving its ability to document-level understanding and generation (as shown in Figure 1).

In this paper, we are particularly interested in how LLMs such as ChatGPT perform for modeling document-level text, encompassing discourse phenomena such as entity consistency, referential expressions, and coherence. Taking document-level MT as a testbed, we conduct an empirical study from three in-depth perspectives:

- **Effects of Context-Aware Prompts**: ChatGPT needs a prompt as guidance to trigger its translation ability. Thus, we enable prompts to guide ChatGPT to consider document-level contexts as long as possible. Jiao et al. (2023) has found that the candidate prompts generally work well and show minor performance differences on sentence-level translation. In this work, we further investigate the effects of prompts on the translation quality and specific discourse phenomena.

- **Comparison of Advanced Translation Models**: While ChatGPT has demonstrated remarkable abilities in long-text NLP tasks, we are specifically interested in how it performs on document-level translation. Consequently, we conduct a systematic comparison of commercial MT products and advanced document-level approaches, utilizing both automatic and human evaluations to assess their discourse awareness.

- **Analysis of Discourse Modelling Abilities**: A more challenging question is the extent to which ChatGPT capture and utilize discourse knowledge. To answer this question, we introduce a probing method through contrastive testing and explanation. In addition, the impact of various training techniques on the ability of LLMs to model discourse has not been thoroughly investigated. We compare variant models of ChatGPT that incorporate techniques such as code pretraining, supervised fine-tuning (SFT), and reinforcement learning from human feedback (RLHF). However, this is not a strict comparison because there are other confounding variables employed during the evolution of ChatGPT. In general, we hope to pose this open question that stimulates reflection and sparks further investigation.

We conduct experiments on a variety of document-level translation benchmarks, covering three language pairs (i.e. Chinese⇒English,

English⇒German and English⇒Russian) and seven domains (i.e. news, social, fiction, Q&A, TED, Europarl, and subtitle). We adopt a comprehensive set of evaluation methods to measure the performance of the models on document-level translation, including general-purpose metrics, discourse-specific metrics, and human evaluation. The **main contributions** are:

- Our empirical study shows the superior capabilities of LLMs over advanced MT systems and methods on document-level translation, indicating their potential to form a new paradigm.

- We establish a benchmark with a probing method to thoroughly assess the document-level translation quality and the ability of learning discourse knowledge, which will be made available for future research.

- To facilitate future research on document MT, we publicly release the instruction-based benchmark, system outputs as well as human annotations.

## 2 Experimental Step

### 2.1 Dataset

Table 1 shows statistics of document-level datasets used in our experiments. About Group #1, we utilized the latest datasets, mZPRT (Wang et al., 2022) and WMT2022 (Kocmi et al., 2022), for evaluation to ensure that the testing data had not been used in commercial systems (e.g. Google Translate and ChatGPT) before. As seen, this covers four domains (i.e. news, social media, web fiction, and Q&A forum) in Chinese⇒English. Regarding Group #2, we utilized four widely-used benchmarks to compare established document-level methods with GPT-like applications. This covers three domains (i.e. TED talks, news commentary, European Parliament) in Chinese⇒English and English⇒German. In Group #3, we employed an English⇒Russian contrastive testset (Voita et al., 2019b) that specifically targets discourse phenomena, such as deixis, ellipsis, and lexical cohesion. We use this dataset to further exploit models' capacity for modeling discourse knowledge.

As seen, we also report the average length of a document ($|W|/|D|$), which can be considered a measure of the complexity of discourse modeling. As the length of the document increases, it becomes more challenging to model accurate cohesive devices and discourse structure. From this perspective, the mZPRT Fiction and IWSLT TED datasets pose a greater challenge compared to others.

| ID | Domain | Source | Language | \|D\| | \|S\| | \|W\| | \|W\|/\|D\| |
|----|--------|--------|----------|------|------|------|-------------|
| 1 | News | WMT2022 | Zh⇒En | 38 | 505 | 16.1K/18.5K | 424 |
| | Social | | | 25 | 478 | 16.4K/13.3K | 656 |
| | Fiction | mZPRT | Zh⇒En | 12 | 857 | 17.1K/16.6K | 1,425 |
| | Q&A | | | 182 | 1,171 | 15.0K/22.1K | 82 |
| 2 | TED | IWSLT2015 | Zh⇒En | 62 | 6,047 | 116.9K/101.5K | 1,885 |
| | | IWSLT2017 | En⇒De | 23 | 2,271 | 38.1K/33.8K | 1,657 |
| | News | News Commentary v11 | En⇒De | 155 | 2,999 | 56.8K/53.9K | 366 |
| | Europarl | Europarl v7 | | 360 | 5,134 | 130.1K/120.9K | 361 |
| 3 | Subtitle | OpenSub2018 | En⇒Ru | 6,000 | 24,000 | 187.8K/514.8K | 31 |

Table 1: Statistics of datasets for document-level translation and analysis. We select the four latest benchmarks and four commonly-used ones, covering diverse domains and languages. We count the number of documents |D|, sentences |S|, and words |W| in terms of source/target language. The average length of a document |W|/|D| can be considered a measure of the complexity of discourse modeling. Note taht, WMT2022 includes two references, whereas others have only one.

**Discussion on Data Contamination** We conducted ChatGPT in March 28∼31 2023 with the official notice "the training data is up to September 2021".[4] Different from previous work that leaned on dated or single-type datasets to assess ChatGPT's capabilities (Jiao et al., 2023; Lu et al., 2023), we carefully chosen both lastst, public and diverse testsets to mitigate the risks associated with data contamination. Taking Probing Discourse Knowledge (Section 5.1) for example, while the contrastive testset used for the prediction task originated in 2019, the evaluation on the explanation task remains devoid of public references. Our conclusions are comprehensively made by considering both prediction and explanation results, balancing out any potential data contamination concerns. Despite precautions, there remains a risk of data contamination, given that publicly available datasets are easily incorporated into LLM training (e.g. pretraining, SFT, or RLHF). A better way is to consistently integrate and employ the latest datasets when evaluating LLMs.

## 2.2 Evaluation Method

**Translation Quality** We evaluate different approaches and systems using classical sentence- and document-level evaluation metrics. About sentence-level metrics, we employ the commonly-used sacreBLEU (Post, 2018) and TER (Snover et al., 2006). Additionally, we utilize COMET (Rei et al., 2020), which leverages pretrained language

models to achieve high correlations with human quality judgments. About document-level metrics, we report document-level sacreBLEU (d-BLEU) (Liu et al., 2020), which is computed by matching n-grams in the whole document. Note that all evaluations are case-sensitive. To facilitate sentence-level evaluation of document outputs, we implemented automatic sentence splitting/alignment[5] on the output and then manually corrected errors.

**Discourse Awareness** To target specific discourse phenomena, we utilized two targeted metrics, namely, CTT and AZPT, which respectively evaluate the consistency of terminology translation and accuracy of zero pronoun (ZP) translation. Regarding CTT, one repeated terminology should keep the same translation throughout the whole document (Xiao et al., 2011). We adopt a lexical translation consistency metric (Lyu et al., 2021):

$$\mathrm{CTT} = \frac{\sum_{t \in \mathbf{TT}} \frac{\sum_{i=1}^{k} \sum_{j=i+1}^{k} \mathbb{1}(t_i = t_j)}{C_k^2}}{|\mathbf{TT}|} \quad (1)$$

for each terminology word $w \in TT$, the $C_k^2$ denotes the size of the combination of translation set $(t_1, \ldots, t_k)$, and function $\mathbb{1}(t_i = t_j)$ returns 1 if $t_i$ is same as $t_j$, otherwise 0. The metric illustrates how frequently translation pairs of $w$ is same within a document. The higher the metric value is, the more likely $w$ is translated consistently. Regarding ZP, it is a discourse phenomenon that appears frequently in pronoun-dropping (pro-drop)

---

[4]https://platform.openai.com/docs/models/gpt-4.

[5]https://github.com/rsennrich/Bleualign.

| ID | BLEU↑ | | TER↓ | | COMET↑ | | d-BLEU↑ | | cTT↑ | AZPT↑ |
|---|---|---|---|---|---|---|---|---|---|---|
| | News | Fiction | News | Fiction | News | Fiction | News | Fiction | Fiction | Fiction |
| **Base** | 25.5 | 12.4 | 62.7 | 85.0 | 0.420 | 0.095 | 28.2 | 15.4 | 0.19 | 0.39 |
| **P1** | 25.8 | 13.9 | 63.8 | 82.8 | **0.483** | 0.124 | 28.7 | 17.0 | 0.29 | 0.41 |
| **P2** | 26.2 | 13.8 | 61.4 | 81.8 | 0.479 | **0.155** | 28.8 | 16.5 | 0.28 | 0.41 |
| **P3** | **26.5** | **14.4** | **61.1** | **81.1** | 0.469 | 0.154 | **29.1** | **17.4** | **0.33** | **0.44** |

Table 2: Ablation study of document-level prompts (detailed in Table 3) on Chinese⇒English datasets using ChatGPT. We use BLEU and d-BLEU to measure sentence- and document-level translation quality. We also conduct two targeted evaluation metrics to measure specific discourse phenomena: accuracy of zero pronoun translation (AZPT) and consistency of terminology translation (cTT). Base is a sentence-level baseline system using InstructGPT API without any context-based chat box.

| ID | Prompt |
|---|---|
| **P1** | Please provide the TGT translation for the sentence: **S** |
| **P2** | Translate the following SRC sentences into TGT: [**S**$_1$], [**S**$_2$] . . . |
| **P3** | (Continue) Translate this document from SRC to TGT: **S**$_1$ **S**$_2$ . . . |

Table 3: The prompts suggested by ChatGPT for document-level translation. SRC and TGT denote source and target languages, respectively. Each document is orderly processed in one "Chat Box" while each prompt is fed into one "Conversational Turn". P1 represents separately translating each sentence **S** in a document. P2 or P3 means translating a document w/wo a sentential boundary tag "[]".

languages such as Chinese and Japanese. Recovering ZPs in a target language (non-pro-drop) needs an understanding of the discourse structure. We used the AZPT score to measure the accuracy of ZP translation (Wang et al., 2022):

$$\text{AZPT} = \frac{\sum_{z \in \mathbf{ZP}} A(t_z|z)}{|\mathbf{ZP}|} \quad (2)$$

where **ZP** is the list of zero pronouns in the source sentences, $t_z$ is the generated translation for the zero pronoun $z$, and $A(t_z|z)$ is a binary scorer to judge whether $t_z$ is the correct translation of $z$.

**Human Evaluation**  To thoroughly validate our conclusions, we also conduct a human evaluation (see Section Limitation.). We establish two sets of evaluation criteria: 1) *general quality*, covering aspects such as fluency and adequacy; 2) *discourse-aware quality*, including factors such as consistency, word choice, and anaphora. The detailed scoring criteria are listed in Appendix§ A.2. Ac-

cordingly, each output will be assigned two distinct scores (0∼5). For each domain subset, we assessed 100 instances, with each instance containing outputs from 5 different systems. This amounted to an evaluation of roughly 70K words in total. The scores were assigned to each window of neighboring sentences, taking into account the context provided by the entire document. Our intent was for evaluators to consider discourse properties beyond single sentences, while also avoiding the difficult task of evaluating an entire document. We employed two professional evaluators for our study. The payment and background is detailed in Section Ethical Considerations and Appendix§ A.2, respectively. Besides, our annotators were given practice items, and the annotations reaches 0.86 Cohen's kappa scores (McHugh, 2012), demonstrating that the annotators work efficiently and consistently under this guideline.

## 3 Effects of Context-Aware Prompts

### 3.1 Motivation

Existing document NMT methods can be mainly classified into two categories: multi-sentence (Wang et al., 2017; Voita et al., 2018; Tu et al., 2018) and whole-document (Macé and Servan, 2019; Bao et al., 2021) approaches. ChatGPT is capable of not only handling long text in a single conversational turn but also recalling the entire context in the chat box. Accordingly, we design prompts to trigger the document-level translation ability of ChatGPT.

The prompt engineering is necessary to ensure ChatGPT's robust ability to interpret instructions and to model long-term dependencies. Our research confirms the neutrality and representativeness of various prompts, allowing other researchers

| Model | Automatic (d-BLEU) | | | | | Human (General/Discourse) | | | | |
|---|---|---|---|---|---|---|---|---|---|---|
| | News | Social | Fiction | Q&A | Ave. | News | Social | Fiction | Q&A | Ave. |
| Google | 27.7 | 35.4 | 16.0 | 12.0 | 22.8 | 1.9/2.0 | 1.2/1.3 | 2.1/2.4 | 1.5/1.5 | 1.7/1.8 |
| DeepL | **30.3** | 33.4 | 16.1 | 11.9 | 22.9 | 2.2/2.2 | 1.3/1.1 | 2.4/2.6 | 1.6/1.5 | 1.9/1.9 |
| Tencent | 29.3 | **38.8** | **20.7** | 15.0 | **26.0** | 2.3/2.2 | 1.5/1.5 | 2.6/2.8 | 1.8/1.7 | 2.1/2.1 |
| GPT-3.5 | 29.1 | 35.5 | 17.4 | 17.4 | 24.9 | 2.8/2.8 | 2.5/2.7 | **2.8**/**2.9** | 2.9/2.9 | 2.8/2.8 |
| GPT-4 | 29.7 | 34.4 | 18.8 | **19.0** | 25.5 | **3.3**/**3.4** | **2.9**/**2.9** | 2.6/2.8 | **3.1**/**3.2** | **3.0**/**3.1** |

Table 4: Comparison between commercial MT systems and LLM applications on Chinese⇒English datasets using both automatic and human evaluation methods. The human evaluation based on a scale from 0∼5 encompasses two dimensions: general quality and discourse awareness (detailed in Table 12). The significant test is detailed in Appendix §A.1.

to utilize them with confidence, unburdened by concerns of unintended biases.

## 3.2 Comparison of Different Prompts

We query ChatGPT itself for advice and obtain a number of candidate prompts, and then refine them into three document-level prompts as shown in Table 3. We utilize P1 to translate a document sentence by sentence, with each sentence placed in a single conversational turn and the entire document contained within one chat box. This mainly takes advantage of ChatGPT's long-term modeling ability in the chat box. P2 and P3 combine multiple continuous sentences and translate them in one conversational turn until the entire document is finished. This aims to maximize the length of document as input. The only difference is whether or not the sentential boundary tag "[]" is inserted into each sentence.

We compare the three candidate prompts on the Zh⇒En translation task using two testsets, WMT2022 News and mZPRT Fiction. Table 2 shows the translation quality in terms of a variety of automatic evaluation metrics. In general, ChatGPT reliably performs well with three candidate prompts, showing only minor variations in performance. This aligns with prior findings in sentence-level translation with ChatGPT (Jiao et al., 2023). Out of the three prompts, the prompt involved multi-turn contexts without sentence boundaries (P3) achieves the best scores in most evaluation metrics, except for COMET. Regarding discourse phenomena, P3 outperforms other candidates with better consistency of terminology translation and higher accuracy of ZP translation. Upon examining the output samples, we noticed that ChatGPT may sometimes forget the sentential boundary tag

of P2 and combine all sentences together. **Takeaway:** (1) *Despite translating a document sentence by sentence, ChatGPT's ability to model long-term dependencies already exists within the chat box.* (2) *Increasing document length as a input can further enhance translation quality and discourse awareness.* (3) *ChatGPT tends to translate a document without adhering to strict sentential boundaries, mirroring a natural approach adopted by humans during document translation, which doesn't necessitate sentence-to-sentence translation.*

## 4 Comparison of Translation Models

In this section, we compare various systems and methods for the document-level translation task. In the following experiments, we use the P3 prompt for ChatGPT and the same document-level window size for MT models as the default setting.

## 4.1 ChatGPT vs. Commercial Systems

Commercial systems are known for their high accuracy and efficiency in translation, making them a strong contender for any machine translation evaluation. By comparing with commercial systems, we can gauge ChatGPT's performance relative to the best available MT technologies. We compare GPT-3.5/GPT-4 with three commercial translation products, including Google Translate,[6] DeepL Translate,[7] and Tencent TranSmart (Huang et al., 2021).[8] We employ both automatic (d-BLEU) and human evaluation (general/discourse-aware quality) as detailed in Section 2.2.

Table 4 shows the results. When evaluated using d-BLEU, commercial MT systems generally out-

---
[6] https://translate.google.com.
[7] https://www.deepl.com.
[8] https://transmart.qq.com.

| Model | ZH⇒EN | | EN⇒DE | | | | | |
| | TED | | TED | | News | | Europarl | |
| | BLEU | d-BLEU | BLEU | d-BLEU | BLEU | d-BLEU | BLEU | d-BLEU |
|---|---|---|---|---|---|---|---|---|
| MCN | 19.1 | 25.7 | 25.1 | 29.1 | 24.9 | 27.0 | 30.4 | 32.6 |
| G-Trans | - | - | 25.1 | 27.2 | 25.5 | 27.1 | 32.4 | 34.1 |
| Sent2Sent | 19.2 | 25.8 | 25.2 | 29.2 | 25.0 | 27.0 | 31.7 | 33.8 |
| MR-Doc2Sent | 19.4 | 25.8 | 25.2 | 29.2 | 25.0 | 26.7 | 32.1 | 34.2 |
| MR-Doc2Doc | - | 25.9 | - | 29.3 | - | 26.7 | - | 34.5 |
| Sent2Sent⋆ | 21.9 | 27.9 | 27.1 | 30.7 | 27.9 | 29.4 | 32.1 | 34.2 |
| MR-Doc2Sent⋆ | 22.0 | 28.1 | 27.3 | 31.0 | 29.5 | 31.2 | 32.4 | 34.5 |
| MR-Doc2Doc⋆ | - | **28.4** | - | 31.4 | - | 32.6 | - | **34.9** |
| ChatGPT | - | 28.3 | - | **33.6** | - | **39.4** | - | 30.4 |

Table 5: Comparison between document-level NMT methods and LLM applications on Chinese⇒English and English⇒German benchmarks using commonly-used BLEU and d-BLEU metrics. "⋆" indicates using additional sentence-level corpus for model pre-training.

perform LLM-based systems, except for the Q&A domain, which involves informal spoken language. While the difference in performance is not significant in the news domain (e.g. the gap between DeepL and GPT-4 is only 0.6 points), it is considerable in the social media and web fiction domains (i.e. the gaps are 3.3 and 1.9 points). A surprising finding is that GPT-4 and GPT-3.5 perform significantly better than MT systems in terms of human evaluation. The potential reasons may be: (1) d-BLEU only measures the similarity of the n-grams between the MT output and the reference translations. However, human takes into account additional factors such as coherence, fluency, and naturalness of the translation, which may not necessarily correlate with d-BLEU scores. (2) ChatGPT and MT systems may have different strengths and weaknesses. For example, ChatGPT may be better at modeling long-term dependencies and capturing discourse-level information, which could result in higher human evaluation. On the other hand, MT systems may perform better in terms of word-level accuracy, which is reflected in d-BLEU. Note that, our findings is distinguished from Neubig and He (2023). Focusing on long-text translation, we compare ChatGPT with MT systems, and underscore ChatGPT's enhanced capacity to model long-term dependencies in comparison to MT systems. On the other hand, Neubig and He (2023) investigate the varying performances of GPT models based on sentence length. They found that GPT models perform better on shorter sentences while worse on longer ones. Karpinska and Iyyer (2023) recently

highlighted that GPT-3.5 has the capability to utilize document-level context effectively for literary translation, yet it is not free from critical errors. While the Fiction testset in our work is categorized under literary, we did not find obvious omission errors in the output. A more detailed comparison is earmarked for future exploration. Karpinska and Iyyer (2023) recently pointed that GPT-3.5 can effectively leverage document-level context for literary translation, but critical errors persist. Although the Fiction subset belongs to literary, we did not find omission errors in the output and we leave this fine-grained comparison for future work. **Takeaway:** (1) *There is a certain degree of discrepancy discrepancy between human and automatic evaluation, which potentially provide complementary reference points when measuring the quality of document-level translations*; (2) *This discrepancy underlines the complexity inherent in accurately evaluating the capabilities of such systems. We further explore evaluation methods in Section 5.1*.

### 4.2 ChatGPT vs. Document NMT Methods

Document NMT methods are specifically designed to handle part or entire documents, making them a relevant point of comparison for evaluating Chat-GPT's ability to model long-term dependencies and discourse phenomena. We compare with five advanced document-level NMT models:

- **MCN** (Zheng et al., 2020): A multi-channel network that integrates a hierarchical encoder and a parallel decoder, which leverages the document structure and semantics for translation.

| ID | Prompt |
|----|--------|
| **P4** | Given an SRC text:{**D**}. Which one is the correct TGT translation as follows: [**T**$_1$], ..., [**T**$_m$]. **Why?** |

Table 6: The prompt for probing discourse knowledge encoded in LLMs. SRC and TGT denote source and target languages, respectively. **D** represents a document contains several sentences. **T**$_1$ ... **T**$_m$ refer to the translation candidates, where only one of them is a positive translation and the others are negative due to the modification of discourse-specific words.

- **G-Trans** (Bao et al., 2021): A graph-based transformer that incorporates document-level discourse structure as a directed acyclic graph, enhancing the representation of the context.
- **Sent2Sent**: A superior sentence-level baseline that employs a transformer architecture to translate each sentence independently and then merges the translations into a document-level output.
- **MR-Doc2Doc** and **MR-Doc2Sent**: Sun et al. (2022) explore to resolve document translation with the end-to-end, namely document-to-document (Doc2Doc) pattern, and utilize *Multi-resolutional Training*, which combines documents with shorter segments like sentences or paragraphs to improve translation quality (denoted as MR-Doc2Doc). Additionally, they reproduce the document-to-sentence baseline (MR-Doc2Sent) that introduces extra model modules to capture contextual information.

To enable a fair comparison with previous work, we use four widely used document-level translation benchmarks: TED (ZH-EN and EN-DE), News (EN-DE), and Europarl (EN-DE). We adopt tokenized case-insensitive BLEU and d-BLEU as the evaluation metrics. As MR-Doc2Doc and ChatGPT generate document-level translations that are difficult to separate into individual sentences, we only report d-BLEU scores for these models.

Table 5 lists the results. The MR-Doc2Doc with extra model pre-training achieves the best document-level performance among previous models. Thanks to the document-level LM pre-training, ChatGPT easily outperforms MR-Doc2Doc$^\star$ on TED (EN-DE) and News (EN-DE) datasets, obtaining similar performance on TED (ZH-EN) dataset. Surprisingly, ChatGPT performs poorly on the Europarl (EN-DE) dataset, even worse than Sent2Sent. We suspect this phenomenon may be caused by

| Model | deixis | lex.c | ell.infl | ell.VP |
|-------|--------|-------|----------|--------|
| Sent2Sent | 51.1 | 45.6 | 55.4 | 27.4 |
| MR-Doc2Doc | 64.7 | 46.3 | 65.9 | 53.0 |
| CADec | 81.6 | 58.1 | 72.2 | 80.0 |
| DocRepair | **91.8** | **80.6** | **86.4** | 75.2 |
| GPT-3.5 | 57.9 | 44.4 | 75.0 | 71.6 |
| GPT-4 | *85.9 | *72.4 | 69.8 | ***81.4** |

Table 7: Accuracy [%] of translation prediction for specific contextual phenomena (deixis, lexical consistency, ellipsis (inflection), and VP ellipsis) between different models on the English⇒Russian contrastive testset. "*" indicates a significant difference ($p < 0.001$) between GPT-4 and GPT-3.5.

the domain distribution bias of the training data. Moreover, we find that ChatGPT is unstable, and its translation results sometimes exhibit omissions and obvious copying behaviors. Note that, the commonly-used datasets were created between 2012 and 2017, a time frame that raises the possibility of these datasets being incorporated into the training data of newer language models. **Takeaway:** (1) *ChatGPT has exhibited superior performance and may become a new promising paradigm for document-level NMT*; (2) *It is still debatable whether these benchmarks can be considered as appropriate measures for evaluating document-level translation methods. We advocate for greater transparency from model developers regarding their training datasets. Additionally, this highlights the importance of designing innovative evaluation techniques that can reliably assess model capabilities while sidestepping concerns related to data contamination.*

## 5 Analysis of Large Language Models

We analyze the ability of LLMs to capture discourse knowledge from two perspectives: (1) probing the discourse knowledge encoded in LLMs, and (2) examining the impact of different training techniques on discourse modeling.

### 5.1 Probing Discourse Knowledge in LLM

In order to verify whether LLMs truly learn to utilize the context to resolve discourse inconsistencies, we adopt the contrastive test sets proposed by Voita et al. (2019b). This dataset includes deixis, lexicon consistency, ellipsis (inflection), and ellipsis (verb phrase) for evaluating discourse phenomena in English-Russian translations. Each instance has

| Subset | GPT-3.5 | | | GPT-4 | | |
|---|---|---|---|---|---|---|
| | Prediction | Explanation | $r_\phi$ | Prediction | Explanation | $r_\phi$ |
| deixis | 58.0 | 18.0 | **0.293** | *89.0 | *93.0 | 0.279 |
| lex.c | 42.0 | 11.0 | 0.089 | *72.0 | *86.0 | **0.293** |
| ell.infl | **75.0** | 58.0 | **0.398** | 71.0 | *91.0 | 0.184 |
| ell.VP | 74.0 | 75.0 | 0.448 | **82.0** | *94.0 | **0.539** |

Table 8: Human evaluation results of GPT-3.5 and GPT-4 on contrastive test sets. For each test set, we randomly select 100 examples and ask annotators to assess whether the responses generated by the models include the correct prediction and explanation, respectively. We count the accuracy (%) of prediction and explanation for GPT-3.5 and GPT-4, based on which the Phi coefficient ($r_\phi$) is calculated to measure the association between two binary variables (i.e., prediction and explanation). "*" indicates a significant difference ($p < 0.001$) between GPT-4 and GPT-3.5.

a positive translation and a few negative ones that differ by only one specific word. The goal is to determine if a model is more likely to generate a correct translation than incorrect variations. In this experiment, we compare GPT-3.5/GPT-4 with advanced methods, such as Sent2Sent, MR-Doc2Doc, CADec (Voita et al., 2019b) and DocRepair (Voita et al., 2019a), where CADec and DocRepair introduce context-aware post-editing modules to refine the sentence-level translations. For these baselines, we adopt force decoding to generate scores for all translation candidates in each instance. If the score of the positive translation is the highest, then this instance is counted as correct. For ChatGPT, we query them with the prompt P4 in Table 6 to obtain responses and correspondent explanations for each instance. Then some heuristic rules and manual verification are used to calculate final performance.

**Evaluation on Prediction** As shown in Table 7, GPT-3.5 performs worse than DocRepair (discourse-enhanced method) across all discourse phenomena, with particularly significant gaps present in deixis and lexical consistency tests. These results show that it is difficult to handle deixis and lexical consistency phenomena with large-scale document-level pre-training. GPT-4 exhibits significant improvements in these areas, but it still lags behind DocRepair in deixis, lexical consistency, and ellipsis (inflection) phenomena. **Takeaway:** (1) *GPT-3.5 demonstrates lower accuracy in contrastive prediction compared to conventional translation models, whereas GPT-4 exhibits significant improvement.* (2) *As there is no detailed technical report available for GPT-4, we argue that its significant improvements are likely due to the use of supervised data and RLHF. We further explore this in Section 5.2.*

**Evaluation on Explanation** We conduct human evaluations to assess the quality of LLM-generated explanations. This provides an additional way to explore the discourse knowledge contained within LLMs. As illustrated in Table 8, we randomly select 100 examples for each contrastive test set and request native speakers to evaluate whether the models' responses contain the correct prediction and explanation, respectively. Then the Phi coefficient ($r_\phi$) is further calculated to better measure the correlation between two binary variables (i.e., prediction and explanation). We can observe that the accuracy of explanation is often not reflective of the accuracy of prediction, indicating a mismatch in utilizing discourse knowledge for prediction and explanation. In addition, GPT-3.5 is not good at explaining the reason for selecting the correct translation, while GPT-4 exhibits high performance in this aspect and brings better accuracy of prediction. **Takeaway:** (1) *GPT-4 demonstrates a strong ability to explain discourse knowledge.* (2) *Despite GPT-4's superior performance in prediction and explanation, the correlation between prediction and explanation does not appear to be significantly improved compared to GPT-3.5.*

## 5.2 Potential Impacts of Training Techniques

LLMs have become the foundation of natural language processing research (Brown et al., 2020), with recent advancements such as learning from source code (Chen et al., 2021) and RLHF showing promise in improving LLM performance (Ouyang et al., 2022). To investigate the potential impacts of these approaches on discourse modelling, we conduct experiments on Chinese⇒English Fiction and English⇒Russian datasets using different variants of LLMs trained with distinct techniques (detailed in §A.3). Accordingly, we use P3 and P4 prompts.

| Model | ZH⇒EN | | EN⇒RU | Training Techniques in LLMs |
| --- | --- | --- | --- | --- |
| | Automatic | Human | Probing | |
| GPT-3 | 3.3 | n/a | n/a | |
| *InstructGPT* | | | | |
|   + SFT | 7.1 | n/a | n/a | |
|     + FeedME-1 | 14.1 | 2.2/2.5 | 30.5/28.6 | |
| *CodexGPT* | | | | |
|   + FeedME-2 | 16.1 | 2.2/2.3 | 34.4/30.1 | |
|   + PPO | 17.2 | 2.6/2.7 | 58.0/39.4 | |
| GPT-3.5 | 17.4 | 2.8/2.9 | 62.3/40.5 | |
| GPT-4 | 18.8 | 2.6/2.8 | 78.5/91.0 | |

Table 9: Translation quality on Chinese⇒English Fiction (automatic d-BLEU and human general/discourse) and probing performance on English⇒Russian (constrative prediction and explanation). The figure on the left illustrates LLMs along with their training methods (from ChatGPT reports and API notes). 'n/a' skips low performances.

Table 9 shows the results. When SFT with high-quality demonstration examples, the translation performance can achieve 14.1 d-BLEU, which reaches to an acceptable level (InstructGPT +FeedME-1 vs. +SFT). Moreover, code pretraining can improve the document translation quality by 2 d-BLEU points and the discourse knowldge probing by 4/1.5 (CodexGPT +FeedME-2 vs. InstructGPT +FeedME-1). When further adding PPO method, it outperforms all other combination strategies on translation quality, discourse awareness and discourse knowledge probing (CodexGPT +FeedME-2 +PPO vs. others). This shows that RLHF strongly enhances LLM's capability of translation. Lastly, GPT-3.5 and GPT-4 excel in d-BLEU and human evaluation scores as well as probing performance, demonstrating the importance of contextual information for complex, lengthy document translation. **Takeaway**: (1) *Methods like code pretraining, SFT and RLHF appear to enhance the performance of document translation and discourse modeling; (2) However, it is quite challenging to explore the non-open source systems due to various confounding factors introduced during their development. Therefore, we advocate for organizations like OpenAI to provide greater transparency to aid researchers in such explorations.*

## 6 Conclusion and Future Work

We provide a comprehensive evaluation of LLMs (such as GPT-3.5 and GPT-4) for document-level machine translation. Our evaluation covers three main aspects: (1) the effects of discourse-aware prompts, (2) comparison of advanced translation models, and (3) analysis of discourse modelling abilities. With the release of the GPT-4 model, the discourse-aware performance has been significantly improved, making it a promising paradigm for document-level translation. Despite its prowess in generative tasks, it struggles with discerning subtle distinctions for ranking.

In our future work, we plan to explore more document-level evaluation method (Castilho, 2021; Jiang et al., 2023; Kocmi and Federmann, 2023), latest long-text benchmarks (Wang et al., 2023a; Thai et al., 2022; Wang et al., 2023c) and other MT scenarios (Ghazvininejad et al., 2023; Guerreiro et al., 2023; Lyu et al., 2023). Furthermore, we intend to delve into a more detailed analysis and comparison in future work. For instance, we will employ appropriate significant test methods to account for multiple comparisons (e.g. non-parametric Kruskal-Wallis test, Bonferroni correction) and conduct a power analysis (Card et al., 2020; Graham et al., 2020; Vilar et al., 2022; Hendy et al., 2023). About annotation consistency, we further apply the Krippendorff's alpha coefficient and check the confidence interval (Krippendorff, 2011).

## Limitations

We list the main limitations of this work as follows:

- **Potential Inaccuracy of Conclusions**. Our conclusions are derived from experiments conducted on a limited set of datasets, which may not guarantee accuracy or applicability across all contexts. These limitations might inadvertently introduce

bias or overlook certain phenomena, potentially impacting the comprehensiveness of our findings. In response to this concern, we strive to use an extensive array of the most recent datasets, spanning various language pairs and domains. This broad coverage aims to encompass a wide range of linguistic and contextual variations, thereby enhancing the generalizability of our findings.

- **Model Updates in ChatGPT and Reproducibility**. When the underlying model or its parameters of ChatGPT are updated or changed, the conclusions derived from prior evaluations may no longer be valid or entirely accurate. To mitigate this issue, this paper has tried utmost to ensure the reproducibility of our findings: (1) We release all system outputs accompanied by exact timestamps and change logs. This ensures that researchers can reliably reproduce and validate our results. (2) We evaluated all systems at two distinct points: in March and August 2023. While there were minor variations in the exact performance figures between these two evaluations, our overarching conclusions and core findings remained unchanged and consistent.

- **Criteria of Human Evaluation and Refinement**. The design of the criteria still has room for improvement. For example, in the "Discourse Awareness" category, there is only a slight difference between Scores 5 and 4, but a more significant gap between Scores 3 and 2. Given the scarcity of benchmark standards on discourse evaluation from past research, we published the detailed scores for further analysis and highlight this area as an opportunity for refinement.

## Ethical Considerations

- **Annotation Process and Annotator Profile**. A one-week trial annotation phase was conducted for bidding (five companies participated) on our enterprise-level annotation platform. The authors answered questions posted by the annotators of these companies and updated annotation guidelines accordingly. The Q&A history is recorded and updated in the formal annotation phase. After evaluating the trial annotations based on accuracy and consistency, we selected a professional language service company (large enterprise[9]) headquartered in Beijing, China. Their annotators were experts in both source and target languages,

with a background in translation and linguistics (detailed in Table11). To understand any potential biases, annotators were inquired about their familiarity with the specific translation models under evaluation and any affiliations with AI or translation companies. We ensured that none of the annotators had conflicts of interest, and their annotations were routinely cross-checked for consistency. In terms of compensation, annotators received an hourly wage of $37.4. This rate aligns closely with the mean hourly wages observed for U.S. interpreters/translators and foreign language teachers.[10]

- **Annotator Consent and IRB Review**. Prior to the commencement of the study, all participating annotators gave their informed consent, confirming their understanding of the study's objectives and the intended research use of their annotations. An exhaustive IRB review was undertaken and finalized before the project's onset (IRB Protocol Number: IRB-2023-00067 and Approval Date: 01/12/2023). The protocol employed in this work was approved by the Tencent Institutional Review Board. We remain steadfast in our commitment to uphold and adhere to the institution's highest ethical and professional benchmarks throughout our research endeavors.

- **Reproducibility Challenges and Mitigation Strategies**. The evolving nature of closed commercial platforms indeed presents challenges to the reproducibility of research. As these platforms continue to upgrade and improve, previous versions of models may be retired or modified, which can make replication efforts problematic. To tackle this issue, we have made several additions to our paper: (1) *Documentation of Specifics*: We have included exact versions of models we evaluated, along with the precise date of evaluation and other pertinent details. This allows for a clear record of the conditions under which our study was conducted. (2) *Release of System Outputs*: We release all system outputs, which ensures that researchers can reliably reproduce and validate our results. (3) *Advocacy for Archiving Historical Versions*: We emphasize the importance for both the AI community and commercial organizations to maintain archives of previous model iterations. By doing this, re-

---

[9]It is based on the information provided by https://www.kanzhun.com/firm.

[10]It is based on the information provided by https://www.bls.gov/oes/current/oes273091.htm and https://www.bls.gov/oes/current/oes251124.htm.

searchers can readily access and evaluate past versions, ensuring continuity in analysis even as new model versions emerge.

## Acknowledgements

We are grateful to the anonymous reviewers, area chairs and ethics committee for their insightful comments and suggestions which will serve to improve the paper considerably. Their insights are invaluable, not only in helping us present our findings more cautiously, but also in educating us beyond the scope of this single paper.

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

# A Appendix

## A.1 Significance Testing

**Machine Translation** For automatic evaluations, we used the non-parametric one-tailed Wilcoxon signed-rank test (Woolson, 2007). About results in Table 4, the significance test contrasting GPT-3.5/GPT-4 with others yields a p-value of less than 0.05, indicating they do significantly boosts translation quality. For human evaluation, we employ Welch's t-test (Bl, 1947). Table 10 shows the overall significance test by combining all datasets, and the results for each domain are also consistent.

| p | Automatic | | Human | |
|---|---|---|---|---|
| | GPT-3.5 | GPT-4 | GPT-3.5 | GPT-4 |
| Google | 0.0000 | 0.0000 | 0.0000 | 0.0000 |
| DeepL | 0.0000 | 0.0000 | 0.0000 | 0.0000 |
| Tencent | 0.0309 | 0.0001 | 0.0000 | 0.0000 |
| GPT-3.5 | n/a | 0.0022 | n/a | 0.0028 |
| GPT-4 | n/a | n/a | n/a | n/a |

Table 10: The results of significance test in Table 4.

**Probing Task** We performed a Welch's t-test (Bl, 1947) with unequal variances to verify the significance between GPT-3.5 and GPT-4 in Table 7 and 8. We find that the corresponding two-tailed p-value is smaller than 0.001, which indicates the significance between them.

## A.2 Human Evaluation Guidelines

**Guidelines** Table 12 presents the human evaluation criteria for document-level translation, with scores ranging from 0 to 5. A score of 5 indicates excellent overall translation quality, with no grammatical errors, accurate word choice, consistent key terms, and consistent context and tone throughout the passage. A score of 0 indicates poor overall translation quality, with more than half of the translation being mistranslated or missing, inconsistent key terms, and poor fluency and clarity. In between scores reflect varying degrees of translation quality, with factors such as fluency, accuracy, consistency of key terms, and context and tone consistency affecting the score.

**Human Evaluators** We employed two professional evaluators for our study through Tencent's designated supplier. Table 11 detailed their background related to this task.

| Level | Evaluator A (Zh-En) |
|---|---|
| Position | lecture at an international university |
| Education | Ph.D degree in Translation Studies, international university |
| Certification | CATTI Translation Level 1 |
| Experience | Translator for the academic journal; Participant in the Construction and Application Research of the bilingual terminology knowledge base, a National Social Science Fund project. |
| **Level** | **Evaluator B (Zh-En)** |
| Position | manager of quality control at a famous translation company |
| Education | Master in English, international university |
| Certification | TEM8 |
| Experience | Translator for the National Internet Information Center; Translator and proofreader for top company. |
| **Level** | **Evaluator C (Ru-En)** |
| Position | interpreter at an import&export trading company |
| Education | Master in Russian Written Translation, international university |
| Certification | Russian Professional Level 8, English CET6 |
| Experience | Interpreter in several import&export trading companies. |
| **Level** | **Evaluator D (Ru-En)** |
| Position | translator at a translation company |
| Education | Master in Russian Written Translation, international university |
| Certification | Russian Professional Level 8, English CET6 |
| Experience | Work in several translation companies. |

Table 11: The basic background of human annotators.

## A.3 Training Methods in LLMs

- **GPT-3** (Brown et al., 2020): A LLM with 175B parameters pre-trained on large-scale web corpora (approximately 400B tokens) We used OpenAI API davinci.

- InstructGPT (**SFT**) (Ouyang et al., 2022): A GPT-3 model trained with supervised fine-tuning on human demonstrations similar to Ouyang et al. (2022).[11]

- InstructGPT (**FeedME-1**) (Ouyang et al., 2022): An improved version of GPT-3 with supervised

---

[11] https://platform.openai.com/docs/model-index-for-researchers.

fine-tuning on human-written demonstrations and model-generated examples rated by humans with high quality.

- InstructGPT (**FeedME-2**) (Ouyang et al., 2022): An improved version of Codex (Chen et al., 2021) with supervised fine-tuning on human-written demonstrations and human-rated examples with high quality.[12]

- InstructGPT (**PPO**) (Ouyang et al., 2022): An improved version of InstructGPT (FeedME-2) with extra training of RLHF, which is trained with a reward model learned from human comparisons.[13]

- **ChatGPT**: A further improved version of InstrucGPT that can perform tasks via dialogue with users, which is able to take contextual information in dialogue into consideration.

---

[12]https://openai.com/blog/openai-codex.
[13]https://openai.com/blog/
instruction-following.

| Score | General Quality | Discourse Awareness |
|---|---|---|
| 5 | Translation passes quality control; the overall translation is excellent. Translation is very fluent with no grammatical errors and has been localized to fit target language. Word choice is accurate with no mistranslations. The translation is a 100% true to the source text. | No inconsistency relating to key terms such as names, organization, etc. Linking words or expressions between sentences keeps the logic and language of the passage clear and fluent. Context and tone are consistent throughout. The style of the text conforms to the culture and habit of the target language. |
| 4 | Translation passes quality control; the overall translation is very good. Translation is fluent. Any errors that may be present does not affect the meaning or comprehension of the text. Most word choice is accurate, but some may cause ambiguity. Key terms are consistent. Inconsistency is limited to non-key terms. | Logical and language is clear and fluent. Some sentences lack transition but does not affect contextual comprehension. Topic is consistent. Tone and word choice may be inconsistent, but comprehension is not affected. Translation conforms to the culture and habit. |
| 3 | Translation passes quality control; the overall translation is ok. Translation is mostly fluent but there are many sections that require rereading due to language usage. Some word choice is inaccurate or errors but meaning of the sentence can be inferred from context. | Some key terms may be inconsistent. Most sentences translation smoothly and logically but some sentences that may seem abrupt due to lack of linkage. Topic is consistent. Tone and word choice is inconsistent, noticeably affecting the accuracy of reading comprehension. |
| 2 | Translation does not pass quality control; the overall translation is poor. Meaning is unclear or disjointed. Even with multiple rereading, passage may still be incomprehensible. Translation is not accurate to the source text or is missing in large quantities, causing the translation to deviate from the source text. | Many key terms are inconsistent, needing multiple rereading to understand context of the passage. Some linkages are present but overall, the passage lacks fluency and clarity, causing trouble with comprehension. The topic or tone is different from the other passages, affecting reading comprehension. |
| 1 | Translation does not pass quality control; the overall translation is very poor. More than half of the translation is mistranslated or missing. | Key terms are inconsistent, causing great trouble with comprehension. Some linkages are present but overall, the passage lacks fluency and clarity, heavily interfering with comprehension. The topic or tone is different from the other passages, heavily interfering with comprehension. |
| 0 | Translation output is unrelated to the source text. | Output is unrelated to previous or following sections. |

Table 12: Human evaluation criteria on document-level translation.