# OpenReview forum: "Document-Level Machine Translation with Large Language Models"
_EMNLP/2023/Conference — EMNLP 2023 Main_

### Official Review · Reviewer_Dtpc · 2023-08-02

**Soundness:** 4

**Ethical Concerns:**

Yes

**Excitement:**

4: Strong: This paper deepens the understanding of some phenomenon or lowers the barriers to an existing research direction.

**Justification For Ethical Concerns:**

No important details were reported about the human evaluation. We don't know how much the human evaluators were paid, whether the study was IRB-reviewed, who they were or whether the provided a consent to take part in this study.

**Missing References:**

Several papers have explored the capabilities of LLM in terms of MT. At least the following two should be mentioned as they directly relate to the research done by the authors:

(1) Katherine Thai, Marzena Karpinska, Kalpesh Krishna, Bill Ray, Moira Inghilleri, John Wieting, and Mohit Iyyer. 2022. Exploring document-level literary machine translation with parallel paragraphs from world literature. (paragraph-level post-editing with LLMs)

(2) Amr Hendy, Mohamed Abdelrehim, Amr Sharaf, Vikas Raunak, Mohamed Gabr, Hitokazu Matsushita, Young Jin Kim, Mohamed Afify, and Hany Hassan Awadalla. 2023. How Good Are GPT Models at Machine Translation? A Comprehensive Evaluation. (includes document level MT in a setup that resembles P2 setup from this study).

Some other papers would include: Vilar et al., 2022; Guerreiro et al., 2023, Kocmi and Federmann, 2023 (this is about mt-eval with llms)

**Paper Topic And Main Contributions:**

In this paper, the authors investigate the capabilities of large language models, namely GPT-3.5-turbo (chatGPT) and GPT-4, to leverage contextual information. Using a document-level translation task as a testbed, the authors compare the performance of GPT-3.5 and GPT-4 against (1) other commercial systems (GoogleTranslate, DeepL, and Tencent's MT), and (2) publicly available MT systems capable of translating entire documents. The authors study three translation directions: Chinese to English, English to German, and English to Russian.

The authors employ document-level BLEU (d-BLEU), TER, and COMET scores to evaluated the outputs, with d-BLEU being their main choice due to the lack for sentence-to-sentence alignments. They supplement these with a simplified human evaluation, based on two properties rated on a 6-point scale ("discourse awareness" and "overall quality"). In addition, the authors examine the ability of large language models to distinguish correct from incorrect translations, specifically where context is crucial (zero-pronouns, discourse deixis, etc.), using a dataset with contextual errors where there is one correct and multiple incorrect translations. The authors also claim that they investigate the effects of different training methods on the model's ability to exploit discourse-level information.

The three main experiments are:

(1) "Discourse-aware prompts" -- Here the authors use 3 prompts (sentence by sentence in chat conversation, entire text with marked sentences, and just the entire text) in order to choose which one works best. The output is evaluated using automatic metrics (BLEU, d-BLEU, COMET, and TER).

(2a) Comparison of chatGPT/GPT-4 with commercial models -- Here the authors employ the best prompt for GPT-3.5-turbo and GPT-4 and evaluate their performance against three commercial systems (GoogleTranslate, DeepL, Tencent TranSmark). For this experiment two datasets published in 2022 are being used. The outputs are evaluated with d-BLEU and simplified human evaluation (6-point scale);

(2b) Comparison of chatGPT/GPT-4 with document-level NMT models -- Here the authors compare the GPT models with five document-level NMT models. The models are tested on datasets which were published over the years the newest being from 2018. The outputs are evaluated using BLEU and d-BLEU.

(3a) Analyzing the capabilities of LLMs to process discourse-level information -- Here the authors employ an existing dataset with translations containing contextual errors published in 2019. The task is to ultimately favor the correct translation over multiple incorrect candidates. The two GPT models are compared with NMT models. In case of GPT models, the authors prompt the models to indicate the correct translation and produce an explanation as to why it is correct. For the other models the authors use forced decoding. The authors report accuracy for each of these models. For the GPT models the authors also report human evaluation of 100 examples where both the choice and explanation were assigned a good or bad label (binary choice).

(3b) Effect of training methods on the ability to model/process contextual information -- Here the authors test different GPTs models (including InstructGPT and Codex) on the translation task. It appears that the datasets used are from 2022 and 2018 but I believe this is not made clear. The authors then use d-BLEU and simplified human evaluation (6-point scale).

**Questions For The Authors:**

Question A: Did you perform any significance testing, visualize the scores, inspect the trends, and analyze the data and its distribution beyond looking at the  average scores?

Question B: How many evaluators were employed? How much were they compensated? How many documents were evaluated (only for one task it is mentioned that 100 were sampled)? Was there only one rating per document? Who were these people?

Question C: In line 361 you write that it is debatable whether these benchmarks can be considered appropriate, I agree, but I don't see this discussed anywhere apart from takeaways.

Question D: How were COMET/BLEU/TER scores computed for P3 in table 2? Since later for this prompt only d-BLEU is used due to the lack of sent-to-sent alignments.

**Reasons To Accept:**

- The authors explore an important and very timely task of document-level translation,
- The authors add human evaluation (though details are missing and data analysis is problematic),
- The authors experiment with many types of texts (news, fiction, Q&A)

**Reasons To Reject:**

UPDATE: The authors addressed most of my concerns however, I believe that the first and second points are still valid and should be discussed as potential limitations (i.e., there are too many confounding variables to claim that one is investigating an impact of different training methods; and the datasets might have been - and probably were - used for RLHF).

UPDATE 2: the concerns were addressed by the authors to the extend it was possible with the current design.

- The authors claim to investigate the effect of different training methods on the processing of discourse-level information (as one of three main experiments), however, it is questionable whether what we see is the effect of different training methods, different training data, or perhaps the data used for RLHF (rather than RLHF alone, that is it is possible that the MT datasets used in this study were used to create RLHF examples). Since the authors research black box models behind an API, I do not think we can make any claims about the effect of the training method (of which we know little) on the model's performance.

- Data contamination might have influenced the evaluation - The authors employ various existing datasets. While two of these datasets do have publication date past the openAI's models' training cutoff point (made public in August 2022), this seems not to be the case for the other datasets employed in this study (including the dataset with contextual errors). It is likely that these were included in the training data of the LLMs being evaluated. Furthermore, with the RLHF models, it is also possible (and quite likely) that MT datasets published post-training were employed to create the RLHF data. For instance the WMT22 dataset was made public in August 2022, which gives companies like OpenAI plenty of time to retrieve it, reformulate it into training examples, and use for RLHF.

- The authors discuss how certain methods are significantly different from others, yet no significance testing is done to support these claims. For example, in line 486 the authors write "The conversational ability of ChatGPT and GPT-4 significantly boosts translation quality and discourse awareness" -- the difference between zh->en ChatGPT (17.4 d-BLEU; 2.8/2.9 humeval) and GPT-4 (18.8 d-BLEU; 2.6/2.8 humeval) and the scores for FeedME-2 (16.1 d-BLEU; 2.2/2.3 humeval) and PPO (17.2 d-BLEU; 2.6/2.7 humeval) is minimal and it's hard to say whether it is significant without proper testing (including checking the distribution and accounting for multiple comparisons).

- The main automatic method used throughout this paper is d-BLEU, which works best with multiple references, yet I believe only one is given. I understand that there are limited automatic options for document level evaluation where the sentences cannot be aligned. Some researchers used sliding windows for COMET, but that's not ideal (yet worth trying?). That is why the human evaluation is so important, and the one in this paper is lacking.

- Human evaluation - many important details are missing so it is hard to judge the research design (more questions below); however what bothers me most is that the authors construct an ordinal scale with a clear cutoff point between 2 and 3 (for general quality especially), yet they present only average scores. I do believe that "5: Translation passes quality control; the overall translation is excellent (...)" minus "4: Translation passes quality control; the overall translation is very good" is not the same "one" as "3: Translation passes quality control; the overall translation is ok. (...)" minus "2: Translation does not pass quality control; the overall translation is poor. (...)". It is clear that the difference between 5 and 4 is minimal, while between 3 and 2 is much bigger. Simple average is not a proper way to analyze this data (a proper analysis would include histograms of scores, possibly a stacked bar with proportion of choices, statistical testing).

- Another issue with the human evaluation is that it appears that the evaluators were asked to evaluate an entire document by assigning it one score. Note that this is a cognitively demanding and difficult task for the evaluators. The results are likely to be unreliable (please see Sheila Castilho's work on this topic). There is also no indication that the annotators were at least given some practice items.


- "Discourse-aware prompts" - I am not sure what this experiment was about. It seems that the idea was to evaluated how the availability of discourse information can improve the translation, but if that is so, then all three setups did have discourse level information (hence this evaluation is impossible). The only thing this seems to be doing is checking in which way the information should be presented (one sentence at a time in a chat, all sentences at once but clearly marked, or the entire document at once without sentence boundaries).

**Reproducibility:**

3: Could reproduce the results with some difficulty. The settings of parameters are underspecified or subjectively determined; the training/evaluation data are not widely available.

**Reviewer Confidence:**

5: Positive that my evaluation is correct. I read the paper very carefully and I am very familiar with related work.

**Typos Grammar Style And Presentation Improvements:**

I believe that we should avoid talking about "asking model", model's ability "for comprehending", and "ability to document-level awareness". This sounds as if the model had ability to comprehend and be aware, while I believe that the model processes text, and leverage information (rather than "is aware" or "understands").

line 299: discrepancy is repeated

If possible, the exact date for when GPT-4 and chatGPT (GPT-3.5-turbo) were used should be provided, given that these two models receive constant updates, sometimes more often than once a month.

Is the GPT-3 model davinci-002? This information seems to be missing.

---

> ### Author Rebuttal · Authors · 2023-08-29
>
> **Q1. About effects of training methods (Section 5.2) and analyzing black box.**
>
> Indeed, we are curious about ChatGPT's evolution in terms of discourse modelling, thus we leave this open question at the end the this paper (Section 5.2). We hope to stimulate further research in areas such as the interplay of code and text in pre-training. A detailed dissection of the impact of individual training factors (e.g. complex interplay between training methods and training data), while valuable, is beyond the scope of this study.
>
> We recognize the intricacies of pinpointing specific causes due to the black-box nature of the models. Thus, this analysis is grounded on observable outcomes, and we aim to provide a comparative perspective rather than absolute claims on the internal workings of the models. Your feedback is invaluable, and we will make these distinctions more evident in the revised version.
>
> **Q2. About data contamination.**
>
> Our experiments on ChatGPT were conducted in March 2023, and the OpenAI's official notice about the date of training data is "Up to Sep 2021" (according to https://platform.openai.com/docs/models/gpt-4). Different from previous work that often leaned on dated or single-type datasets to assess ChatGPT's capabilities [1-2], we have carefully chosen both lastst, public and diverse testsets to mitigate the risks associated with data contamination.
>
> Given the concerns regarding data contamination, we took rigorous steps to ensure the reliability and integrity of our evaluation:
> 1. **Latest Datasets**. Instead of using older testsets, we have carefully chosen the latest document-level translation datasets, specifically those published in 2022, ensuring a gap between publication and the model's training cut-off.
> 2. **Diverse Domains and Languages**. Instead of  relying on single-type testsets, we have consciously chosen eigth datasets covering six domains and two language pairs, offering a holistic view of ChatGPT's performance.
> 3. **Multi-dimensional Evaluation**. Beyond merely assessing translation quality, we have additionally proposed probing task, diving deeper into the discourse knowledge embedded within models.
>
> Taking contrastive testing for example (Section 5.1), while the dataset used for prediction originated in 2019 [3] , the evaluation on explanation remains devoid of public references. This makes it improbable for ChatGPT to have been trained on it. Our conclusions are comprehensively made by considering both Prediction and Explanation Evluation, balancing out any potential data contamination concerns.
>
> [1] Wenxiang Jiao, Wenxuan Wang, Jen-tse Huang, Xing Wang, and Zhaopeng Tu. 2023. Is chatgpt a good translator? a preliminary study.
>
> [2] Hongyuan Lu, Haoyang Huang, Dongdong Zhang, Haoran Yang, Wai Lam, Furu Wei. 2023. Chain-of-Dictionary Prompting Elicits Translation in Large Language Models.
>
> [3] Elena Voita, Rico Sennrich, and Ivan Titov. 2019. When a good translation is wrong in context: Context-aware machine translation improves on deixis, ellipsis, and lexical cohesion.
>
> **Q3. About significance testing.**
>
> We have indeed carried out significance testing for all our results:
>
> 1. **Machine Translation**. For automatic evaluations, we used the non-parametric one-tailed Wilcoxon signed-rank test. Taking results in Table 4 for example, the signficance test contrasting ChatGPT/GPT-4 with others yields a p-value of less than 0.05, indicating they do significantly boosts translation quality. The following table only shows the overall significance test by combining all datasets, and the detailed results for each domain are also consistent. For human evaluation, we will employ statistical power analysis to gauge the reliability and consistency of human judgments.
>
> |       p      |   Google  |   DeepL    |  Tencent   | GPT-3.5 | GPT-4 |
> |-----------|:------------:|:-------------:|:-------------:|:-----------:|:---------:|
> | **GPT-3.5** | 0.000000 | 0.000000 | 0.030863 |     n/a      |    n/a   |
> | **GPT-4**    | 0.000000 | 0.000000 | 0.000148| 0.002180 |    n/a   |
>
> 2. **Probing Task**. We performed a ttest with unequal variances to verify the significance of different methods on deixis and lexical consistency tests, i.e., DocRepair vs. ChatGPT and  ChatGPT vs. GPT-4. We find that the corresponding two-tailed pvalue is smaller than 0.001, which indicates the significance between them.
>
> Our revised version will incorporate heatmaps displaying all significance test outcomes in the Appendix.
>
> [4] Yvette Graham, Barry Haddow, Philipp Koehn. 2022. Statistical Power and Translationese in Machine Translation Evaluation.
>
> **Q4. About lacking of multiple references and human evaluation.**
>
> The official WMT2022 datasets used in our paper indeed contain multiple references, which are associated with News and Social testsets in Table 2 and 4. We will add this details in Section 2.1.
>
> Your observations about the importance of human evaluation resonate with our stance. We noticed that in sections like "Main Contributions" and "Reasons To Accept," our human evaluations were positively acknowledged, whereas, under "Reasons To Reject," there's a mention of an absence of human evaluation ("That is why the human evaluation is so important, and the one in this paper is lacking.").
>
> **Q5. About human evaluation critera and missing details.**
>
> Please see Q8 for more details regarding human evaluators.
>
> We appreciate your pointing out the nuances in the general quality scale. We will follow your suggestions to conduct more detailed analysis instead of relying solely on simple averages. Besides, all translation outputs along with human scores are released, enabling other researchers to further evaluate and analyze them.
>
> It is worth noting that we did not ask evaluators to assign a single score to an entire document. Instead, scores were assigned to each window of neighboring sentences, taking into account the context provided by the entire document. Our intent was for evaluators to consider discourse properties beyond single sentences, while also avoiding the difficult task of evaluating an entire document. We believe our results hold reliability and were not do not contradict with Sheila Castilho's works. Besides, our annotators were given practice items, and the annotations reaches 0.86% kappa scores, demonstrating that the annotators work efficiently and consistently under this guideline.
>
> As we found limited standards on discourse evaluation in previous studies to reference, we would genuinely appreciate more specific recommendations or references concerning evaluation criteria.
>
> **Q6. About discourse-aware prompts.**
>
> This section is not designed for exploiting how the availability of discourse information can improve the translation. Instead. this is a abliation study about effects of different prompt strategies on document-level translation. This kind of abliation study was usually exploited at the beginning of LLM evaluation works like Jiao et al. [1]. However, document-level translation with ChatGPT-like conversational systems is quite different from sentence-level task. Apart from just the instruction content, it is crucial to consider inherent document factors like multi-turn contexts and sentence boundaries.
>
> We reveal that ChatGPT consistently performs well across three candidate prompts (evaluated using two datasets and six evluation metrics). Among them, the prompt (P3) involved multi-turn contexts without sentence boundaries performs the best. It is necessary on this prompt engineering to ensure ChatGPT's robust ability to interpret instructions and to model long-term dependencies. Our findings serve as a testament to the representativeness and neutrality of different prompts, paving the way for future research. Other reseachers can employ these prompts confidently, without concerns over unintended biases, thereby ensuring the objectivity of their findings.
>
> **Q8. About evaluators.**
>
> We engaged two evaluators for our study. The basic information is as follows:
>
> | Evaluator 1 |
> |-------------------------------------------------|
> | Position: lecture at a top university |
> | Education: Ph.D degree in Translation Studies, top university |
> | Certification: CATTI Translation Level 1 |
> | Professional Experiences: Translator for the academic journal; Participant in the Construction and Application Research of the bilingual terminology knowledge base, a National Social Science Fund project. |
> |**Evaluator 2**|
> | Position: manager of quality control at top translation company |
> | Education: Master in English, top university |
> | Certification: TEM8 |
> | Professional Experiences: Translator for the National Internet Information Center; Translator and proofreader for top company |
>
> For each domain subset, we assessed 100 instances, with each instance containing outputs from 5 different systems. This amounted to an evaluation of roughly 70K words in total. While we count instance in sentence, the actual number of documents varied due to the lengths of different domain. Besides, our annotators were given practice items, and the annotations reaches 0.86% kappa scores, demonstrating that the annotators work efficiently and consistently under this guideline. The compensation provided to the evaluators for this task amounted to $5000 US dollars. We will add the above details in the revised version.
>
> **Q9. About whether these benchmarks can be considered appropriate in line 361.**
>
> Indeed, Line 361 alludes to the debatable nature of the appropriateness of these benchmarks, especially in the context of evaluating LLMs like ChatGPT. This concern stems from the potential risk of data contamination, which could render the benchmark results unreliable. Many commonly-used datasets used in Table 5 were created between 2012 and 2017, a time frame that raises the possibility of these datasets being incorporated into the training data of newer language models.
>
> We advocate for greater transparency from model developers regarding their training datasets. Additionally, this highlights the importance of designing innovative evaluation techniques that can reliably assess model capabilities while sidestepping concerns related to data contamination.
>
> We will add such discussions in the revised verson when given more space.
>
> **Q10. How were COMET/BLEU/TER scores computed for P3 in table 2? Since later for this prompt only d-BLEU is used due to the lack of sent-to-sent alignments.**
>
> We post-processed the document translation output for sentence-level evaluation. Specifically, we conduct automatic sentence alignment [5] and then mannually revise alignment errors.
>
> [5] https://github.com/rsennrich/Bleualign.
>
> **Q11. About missing references, typos and others.**
>
> 1. All paper mentioned will be citated and carefully discussed in our revised version.
> 2. I will revise the statements in the revised version: "asking model" to "query model"; "for comprehending" to "for probing"; "ability to document-level awareness"  to "ability to document-level translation".
> 3. I will reivse the date in the footnote 1: All corresponding results were obtained from ChatGPT and GPT-4 in 28~31 March 2023.
> 4. No, the GPT-3 model is davinci.

---

### Official Review · Reviewer_AmZG · 2023-08-05

**Soundness:** 4

**Excitement:**

4: Strong: This paper deepens the understanding of some phenomenon or lowers the barriers to an existing research direction.

**Paper Topic And Main Contributions:**

This paper discusses the suitability of Large Language Models, such as GPT and ChatGPT, when being tasked with document-level machine translation. Among the contributions are effects of discourse-aware prompts, a comparison of LLMs vs other commercial MT systems, and an examination which of the components in the LLM pipeline (finetuning, reward modeling etc.) contribute most to the LLM's ability to translate in a discourse-aware manner.
The languages looked at are Chinese⇒English, English⇒German and English⇒Russian and the test data is from seven domains (i.e. news, social, fiction, Q&A, TED, Europarl, and subtitles).

**Questions For The Authors:**

* are there any insights / changes in scores etc. when you'd exchange the prompts? or asked differently: how long did you "prompt-engineer" and by how much would the results look differently if you were to spend yet more time finding the ideal prompts or if you were not to prompt-engineer at all?

**Reasons To Accept:**

* emergent field, very important measurements and an assessment of a technology that surely will change the "regular MT" research as well as commercial MT model deployment
* thorough experimentation, which all the rights questions asked and addressed

**Reasons To Reject:**

* none, if my questions are addressed

**Reproducibility:**

4: Could mostly reproduce the results, but there may be some variation because of sample variance or minor variations in their interpretation of the protocol or method.

**Reviewer Confidence:**

4: Quite sure. I tried to check the important points carefully. It's unlikely, though conceivable, that I missed something that should affect my ratings.

---

> ### Author Rebuttal · Authors · 2023-08-29
>
> **Q1. How did changing prompts affect scores? How long was spent on prompt engineering? What would be the impact of further refinement or no prompt engineering?**
>
> In Section 3, we explored various prompt strategies specifically to answer the questions above.
>
> Table 2 reveals that ChatGPT consistently performs well across three candidate prompts, with minor variations in performacne (evaluated using two datasets and six evluation metrics). This aligns with prior findings in sentence-level translation with ChatGPT [1]. It is pivotal to mention that our prompt strategies consider factors inherent to document-level translation. We highlited that the prompt involved multi-turn contexts without sentence boundaries (P3) performs the best.
>
> It is necessary for us to spend approximately one weeks on this prompt engineering to ensure ChatGPT's robust ability to interpret instructions and to model long-term dependencies. Our findings serve as a testament to the representativeness and neutrality of different prompts, paving the way for future research. Other reseachers can employ these prompts confidently, without concerns over unintended biases, thereby ensuring the objectivity of their findings.
>
> [1] Wenxiang Jiao, Wenxuan Wang, Jen-tse Huang, Xing Wang, and Zhaopeng Tu. 2023. Is chatgpt a good translator? a preliminary study.

---

### Official Review · Reviewer_GjYL · 2023-08-05

**Soundness:** 4

**Excitement:**

4: Strong: This paper deepens the understanding of some phenomenon or lowers the barriers to an existing research direction.

**Paper Topic And Main Contributions:**

They do the document-level translation by using ChatGPT and GPT-4 and analyze the result thoroughly.  They show the superiorities of LLMs over advanced MT systems and established a benchmark with a probing method to assess the document-level translation quality and the ability of learning discourse knowledge. They also find that code pre-training and RLHF really help to produce coherent and cohesive translations.

**Questions For The Authors:**

Question A
- In ZenoGPT report, they insist that "GPT models are relatively good on shorter sentence and relatively bad on longer sentences.". In this paper, ChatGPT is good at catching long-term dependency. I wonder which one is correct.

**Reasons To Accept:**

* Interesting findings about analysis of discourse modelling abilities
* They do an in-depth comparison between (ChatGPT/GPT-4) and advanced machine translation service


**Reasons To Reject:**

* Difficulties of reproducibility. ChatGPT is constantly updated.

**Reproducibility:**

3: Could reproduce the results with some difficulty. The settings of parameters are underspecified or subjectively determined; the training/evaluation data are not widely available.

**Reviewer Confidence:**

3: Pretty sure, but there's a chance I missed something. Although I have a good feel for this area in general, I did not carefully check the paper's details, e.g., the math, experimental design, or novelty.

---

> ### Author Rebuttal · Authors · 2023-08-29
>
> **Q1. About difficulties of reproducibility because ChatGPT is constantly updated.**
>
> Thank you for highlighting this concern, a challenge inherent to assessing all non-open-sourced systems such as ChatGPT.
>
> This paper has tried utmost to ensure the reproducibility of our findings:
> 1. We release all system outputs accompanied by exact timestamps and change logs. This ensures that researchers can reliably reproduce and validate our results.
> 2. We evaluated all systems at two distinct points: in March and August 2023. While there were minor variations in the exact performance figures between these two evaluations, our overarching conclusions and core findings remained unchanged and consistent. We will report this in our revised version.
>
> **Q2. Different conlusions with ZenoGPT report.**
>
> The two observations are not contradictory.
> 1. In our study, we compare ChatGPT with MT systems, focusing only on long-text translation. Our results underscore ChatGPT's enhanced capacity to model long-term dependencies in comparison to MT systems.
> 2. Conversely, the ZenoGPT report [1] delves into the differential performance of GPT models between shorter and longer sentences. They found that GPT models perform better on shorter sentences while worse on longer ones.
> We will add this discussion in the revised version.
>
> [1] Graham Neubig and Zhiwei He. 2023. Zeno GPT Machine Translation Report.

---

### Meta-Review · Area_Chair_4sU9 · 2023-09-21

**Recommendation:** 5

**Metareview:**

In this work, the authors perform document-level translation using ChatGPT and GPT-4, and present a thorough comparison of these LLMs with state-of-the-art MT systems on a benchmark evaluating translation quality going from Chinese to English, English to German and Englishto Russian. They also probe which parts of the LLM training pipeline (fine-tuning, reward modeling, etc.) contribute most to its document-level translation capabilities and experiment with different discourse-aware prompts to identify what works best.

After detailed discussions with the authors during the rebuttal phase, the reviewers are all unanimous in their ratings for soundness (4/4/4) and excitement (4/4/4). R3 pointed to many updates that the authors should make in their revised draft, including details of human evaluation criteria, citations to relevant references and to be more precise about the impact of different training techniques.

---

### Decision · Program_Chairs · 2023-10-07

**Decision:**

Accept-Main

**Comment:**

In this work, the authors perform document-level translation using ChatGPT and GPT-4, and present a thorough comparison of these LLMs with state-of-the-art MT systems on a benchmark evaluating translation quality going from Chinese to English, English to German and Englishto Russian. They also probe which parts of the LLM training pipeline (fine-tuning, reward modeling, etc.) contribute most to its document-level translation capabilities and experiment with different discourse-aware prompts to identify what works best.

After detailed discussions with the authors during the rebuttal phase, the reviewers are all unanimous in their ratings for soundness (4/4/4) and excitement (4/4/4). R3 pointed to many updates that the authors should make in their revised draft, including details of human evaluation criteria, citations to relevant references and to be more precise about the impact of different training techniques.